# SSW-GAN: Scalable Stage-wise Training of Video GANs

## Abstract

Current state-of-the-art generative models for videos have high computational requirements that impede high resolution generations beyond a few frames. In this work we propose a stage-wise strategy to train Generative Adversarial Networks (GANs) for videos. We decompose the generative process to first produce a downsampled video that is then spatially upscaled and temporally interpolated by subsequent stages. Upsampling stages are applied locally on temporal chunks of previous outputs to manage the computational complexity. Stages are defined as Generative Adversarial Networks, which are trained sequentially and independently. We validate our approach on Kinetics-600 and BDD100K, for which we train a three stage model capable of generating 128x128 videos with 100 frames.

## 1 Introduction

The field of generative modeling has seen rapid developments over the past few years. Current models such as GPT-3 (Brown et al., 2020) or BigGAN (Brock et al., 2018) are capable of generating coherent long paragraphs and detailed high resolution images. Generative models for videos have high memory requirements that quickly scale with the output resolution and length. Prior works have therefore restricted the video dimensions by operating at low spatial resolution or by only considering a small number of frames to generate (Ranzato et al., 2014; Vondrick et al., 2016a; Tulyakov et al., 2018; Kalchbrenner et al., 2017).

In this work we investigate an approach to reduce the computational costs needed to generate long high resolution videos in the context of Generative Adversarial Networks (GANs). Current GAN approaches require large batch sizes and high capacity models (Clark et al., 2019; Brock et al., 2018). We propose to break down the generative process into a set of smaller generative problems or stages, each stage having reduced computational requirements. The first stage produces a downsampled low-resolution video that is then spatially upscaled and temporally interpolated by subsequent upscaling stages. Each stage is modeled as a GAN problem and stages are trained sequentially and independently.

Each stages only considers a lower dimensional view of the video during training. The first stage is trained to produce full-length videos at a reduced spatiotemporal resolution, while the upscaling stages are trained to upsample partial temporal windows on the previous generations. At inference time, the upscaling stages are applied on the full first stage output in a convolutional fashion to generate full resolution videos.

Learning the upscaling stages on local views of the data reduces their computational requirements. By keeping a fixed temporal window size, computational requirements scale only in output resolution, independent of the final video length. However, upsampling stages consider a limited field of view in time, which could negatively impact the temporal consistency of the full-size generation. To address this problem, we rely on the first low-resolution generation to capture long-term temporal information, and use it to condition the upscaling stages. In particular, we introduce a novel matching discriminator that ensures that outputs are grounded to the low-resolution generation.

Our approach, named SSW-GAN, offers a novel way to decompose the training of large GAN models, inspired by other coarse-to-fine methods that have been explored in the context of images and videos (Denton et al., 2015; Karras et al., 2017; Acharya et al., 2018). In contrast to previous methods, we do not train on full resolution inputs in upscaling stages, and instead impose global

temporal consistency through conditioning on the complete but low resolution of the the first stage output. Our model thus provides significant computational savings, which allow for high quality high resolution generators capable of producing hundred of frames.

Our contributions can be summarized as follows:

- We define a stage-wise approach to train GANs for video in which stages are trained sequentially and show that solving this multi-stage problem is equivalent to modeling the joint data probability of samples and their corresponding downsampled views.
- We empirically validate our approach on Kinetics-600 and BDD100K, two large-scale datasets with complex videos in real-world scenarios. Our approach matches or outperforms state-of-art approaches while requiring significantly less computational resources.
- We use our approach to generate videos with 100 frames at high resolutions with high capacity models. To the best of our knowledge, our method is the first one to produce such generations.

## 2 RELATED WORK

Since Ranzato et al. (2014) and Srivastava et al. (2015a) proposed the first video generation models inspired by the language modeling literature, there have been many papers that have proposed different approaches to represent and generate videos (Luc et al., 2017; 2018; Villegas et al., 2017a;b; Xue et al., 2016).

We propose a model for class conditional video generation, and therefore our setup is closely related to stochastic generative video models. Autoregressive models (Larochelle & Murray, 2011; Dinh et al., 2016; Kalchbrenner et al., 2017; Reed et al., 2017; Weissenborn et al., 2020) approximate the joint data distribution in pixel space without introducing latent variables. These models capture complex pixel dependencies without independence assumptions. However, inference in autoregressive models often requires a full model forward pass for each output pixel, making them slow and not scalable to long high resolution videos, with state-of-the-art models requiring multiple minutes to generate a single batch of samples (Weissenborn et al., 2020).

Variational AutoEncoders (VAEs) define latent variable models and use variational inference methods to optimize a lower bound on the empirical data likelihood (Kingma & Welling, 2013; Rezende et al., 2014; Babaeizadeh et al., 2017). Models based on VRNNs (Chung et al., 2015; Denton & Fergus, 2018; Castrejon et al., 2019) use per-frame latent variables and have greater modeling capacity.

Normalizing flows (NFs) define bijective functions that map a probability distribution over a latent variable to a tractable distribution over data (Rezende & Mohamed, 2015; Kingma & Dhariwal, 2018; Kumar et al., 2019). NFs are trained to directly maximize the data likelihood. The main disadvantage of NFs is that their latent dimensionality has to match that of the data, often resulting in slow and memory-intensive models.

Autoregressive models, VAEs and NFs are trained by maximizing the data likelihood (or a bound) under the generative distribution. It has been empirically observed that such models often produce blurry results. Generative Adversarial Networks (GANs) on the other hand optimize a min-max game between a Generator and a Discriminator trained to tell real and generated data apart (Goodfellow et al., 2014). Empirically, GANs usually produce better samples but might suffer from mode collapse. GAN models for video were first proposed in Vondrick et al. (2016b;a); Mathieu et al. (2015). In recent work, SAVP (Lee et al., 2018) proposed to use the VAE-GAN (Larsen et al., 2015) framework for video. TGANv2 (Saito & Saito, 2018) improves upon TGAN (Saito et al., 2017) and proposes a video GAN trained on data windows, similar to our approach. However, unlike TGANv2, our model is composed of multiple stages which are not trained jointly. MoCoGAN (Tulyakov et al., 2018) first introduced a dual discriminator architecture for video, with DVD-GAN (Clark et al., 2019) scaling up this approach to high resolution videos in the wild. DVD-GAN outperforms MoCoGAN and TGANv2, and is arguably the current state-of-the-art in adversarial video generation. We propose a multi-stage generative model approach with each stage defining an adversarial game and a model architecture based upon DVD-GAN. Recent work (Xiong et al., 2018; Zhao et al., 2020) also proposes multi-stage models, but differently from our approach their stages model different semantic aspects of the generation, such as producing a motion outline.

## 3   METHOD

We consider a dataset of videos $(\mathbf{x}_1, ..., \mathbf{x}_n)$ where each $\mathbf{x}_i = (\mathbf{x}_{i;0}, .., \mathbf{x}_{i;T})$ is a sequence of $T$ frames $\mathbf{x}_{i;t} \in \mathbb{R}^{H \times W \times 3}$. Each $\mathbf{x}_i$ comes from a data distribution $p_d$. Our goal is to learn a generative distribution $p_g$ such that $p_g = p_d$.

### 3.1   STAGE-WISE GENERATIVE PROCESS

While our approach can be generalized to multiple stages, for clarity of exposition, we describe here a two-stage version of the model. We begin by introducing a new variable $\mathbf{x}^l = f(\mathbf{x})$ representing a view on the data. Generally, $f$ denotes an operation that produces a view of $\mathbf{x}$ which decreases its dimensionality, and in this work $f$ is a nearest neighbor downsampling operation that reduces the spatiotemporal resolution of $\mathbf{x}$. Our goal is to model the joint probability distribution $p_d(\mathbf{x}, \mathbf{x}^l)$.

We define a generative model that approximates the joint data-view distribution according to the following factorization:

$$p_g(\hat{\mathbf{x}}, \hat{\mathbf{x}}^l) = p_{g_2}(\hat{\mathbf{x}}|\hat{\mathbf{x}}^l)p_{g_1}(\hat{\mathbf{x}}^l). \tag{1}$$

Each $p_{g_i}$ defines a stage in our model. This formulation allows us to decompose the training of the stages into a sequential process.

**Training Stage 1**   We consider the distribution $p_{g_1}$ in eq. 1 and solve a min-max game with the following value function:

$$V_1(G_1, D_1) = \mathbb{E}_{\mathbf{x}^l \sim p_d}[\log(D_1(\mathbf{x}^l))] + \mathbb{E}_{\mathbf{z}_1 \sim p_{z_1}}[\log(1 - D_1(G_1(\mathbf{z}_1)))], \tag{2}$$

where $G_1$ and $D_1$ are the generator/discriminator associated with the first stage and $p_{z_1}$ is a noise distribution. This is the standard GAN objective. As shown in Goodfellow et al. (2014), the min-max game $\min_{G_1} \max_{D_1} V_1(G_1, D_1)$ has a global minimum when $p_{g_1}(\mathbf{x}^l) = p_d(\mathbf{x}^l)$.

**Training Stage 2**   We formulate a min-max game with the following value function:

$$V_2(G_2, D_2) = \mathbb{E}_{\mathbf{x}^l \sim p_d}\mathbb{E}_{\mathbf{x} \sim p_d(.|\mathbf{x}^l)}[\log(D_2(\mathbf{x}, \mathbf{x}^l))] + \mathbb{E}_{\hat{\mathbf{x}}^l \sim p_{g_1}}\mathbb{E}_{\mathbf{z}_2 \sim p_{z_2}}[\log(1 - D_2(G_2(\mathbf{z}_2, \hat{\mathbf{x}}^l), \hat{\mathbf{x}}^l))], \tag{3}$$

where $G_2$ and $D_2$ are the generator and discriminator of the second stage. The min-max game $\min_{G_2} \max_{D_2} V_2(G_2, D_2)$ has a global minimum when the two joint distributions are equal, $p_d(\mathbf{x}, \mathbf{x^l}) = p_{g_2}(\mathbf{x}|\mathbf{x}^l)p_{g_1}(\mathbf{x}^l)$ (Dumoulin et al., 2016; Donahue et al., 2016). It follows that $p_d(\mathbf{x}|\mathbf{x}^l) = p_{g_2}(\mathbf{x}|\mathbf{x}^l)$ when $p_{g_1}(\mathbf{x}^l) = p_d(\mathbf{x}^l)$. This stage only learns the parameters associated with the distribution $p_{g_2}$, as $p_{g_1}$ is trained in the previous stage. However, even if the distribution $p_{g_1}$ does not match exactly the marginal data distribution, our model still aims at learning a distribution $p_{g_2}$ such that $p_{g_2}(\mathbf{x}|\mathbf{x}^l)p_{g_1}(\mathbf{x}^l)$ approximates the joint data distribution. Therefore, our approach finds a generative distribution $p_g$ such that $p_g(\mathbf{x}, \mathbf{x^l}) = p_d(\mathbf{x}, \mathbf{x^l})$.

### 3.2   TRAINING ON LOCAL VIEWS OF THE DATA

We have shown that we can learn the joint data distribution with a stage-wise strategy. However, this approach does not yet have any significant computational benefits. In its current form, the second stage which upscales the low-resolution view $\mathbf{x}^l$ parametrizes a generative distribution over the full resolution video $\mathbf{x}$. We now proceed to describe a locality assumption that allows us to reduce the output dimensionality of the upscaling stage and therefore also reduce its computational complexity.

We decompose a video $\mathbf{x}$ and its view $\mathbf{x^l}$ into a set of corresponding overlapping temporal windows $\mathbf{x} = (\mathbf{x}_{w_1}, \ldots, \mathbf{x}_{w_n})$ and $\mathbf{x}^l = (\mathbf{x}^l_{w_1}, \ldots, \mathbf{x}^l_{w_n})$. We now assume:

$$p(\mathbf{x}_{w_i}) \perp p(\mathbf{x}_{w_j})|\mathbf{x}^l_{w_i}, \forall i, j \ i \neq j \text{ and } p(\mathbf{x}_{w_i}) \perp p(\mathbf{x}^l_{w_j})|\mathbf{x}^l_{w_i}, \forall i, j \ i \neq j \tag{4}$$

We therefore assume that an output window $\mathbf{x}_{w_i}$ is independent of all other data views conditioned on the corresponding low-resolution window $\mathbf{x}^l_{w_i}$. Given this locality assumption, we can rewrite the conditional data and generative distribution as:

$$p_d(\mathbf{x}|\mathbf{x}^l) = \prod_{i=1}^{n} p_d(\mathbf{x}_{w_i}|\mathbf{x}^l_{w_i}) \text{ and } p_{g_2}(\hat{\mathbf{x}}|\hat{\mathbf{x}}^l) = \prod_{i=1}^{n} p_{g_2}(\hat{\mathbf{x}}_{w_i}|\hat{\mathbf{x}}^l_{w_i}), \tag{5}$$

This allows us to rewrite the value function of eq. 3 as:

$$\mathbb{E}_{\mathbf{x}^l \sim p_d} \mathbb{E}_{\mathbf{x}_w \sim p_d(.|\mathbf{x}_w^l)}[\log(D_2(\mathbf{x}_w, \mathbf{x}_w^l))] + \mathbb{E}_{\hat{\mathbf{x}}^l \sim p_{g_1}} \mathbb{E}_{z_2 \sim p_{z_2}}[\rho(1 - D_2(G_2(z_2, \hat{\mathbf{x}}_w^l)), \hat{\mathbf{x}}_w^l))]. \quad (6)$$

Eq. 6 describes the second stage value function in which we now match distributions of uniformly sampled data windows. Because these window sizes are fixed length and smaller than the full video length, the second stage now models a lower dimensional generative distribution.

This assumption introduces a trade-off between the modeling capacity of the second stage and the dimensionality of its outputs. Local upsampling stages might have trouble maintaining temporal consistency, especially for high resolution details and textures that are not fully defined by the low resolution inputs. In practice we define data windows with overlapping frames to aid maintain consistency. This is equivalent to defining a fully convolutional upscaling model with a temporal field of view smaller than the full input sequence. At inference time, we run the model convolutionally over the entire input.

## 4    MODEL ARCHITECTURE

In this section, we describe how we parametrize the different stages of our generative model. A full description is provided in the Appendix.

**Stage 1**    We follow the DVD-GAN architecture (Clark et al., 2019) for the first stage of our model. The DVD-GAN generator stacks units composed by a ConvGRU layer (Ballas et al., 2015), modeling the temporal information, and 2D-ResNet blocks that upsample the spatial resolution. DVD-GAN builds upon the dual discriminator of MoCoGAN (Tulyakov et al., 2018) and defines both a spatial discriminator that randomly samples $k$ full-resolution frames and discriminates them individually, and a temporal discriminator that processes spatially downsampled but full-length videos.

**Upsampling Stage**    The upsampling stage is composed by a conditional generator and three discriminators (spatial, temporal and matching). The conditional generator produces an upscaled version $\hat{\mathbf{x}}_{\mathbf{w}}$ of a low resolution video $\hat{\mathbf{x}}_{\mathbf{w}}^l$. To discriminate samples from real videos, this stage uses a spatial and temporal discriminator, following DVD-GAN. Additionally, we introduce a matching discriminator (MD). The goal of MD is to ensure that the generation $\hat{\mathbf{x}}_{\mathbf{w}}^l$ is a valid upsampling of $\hat{\mathbf{x}}_{\mathbf{w}}$. Conditioning on $\hat{\mathbf{x}}_{\mathbf{w}}^l$ allows to impose global temporal consistency on the generation $\hat{\mathbf{x}}$ at inference time. Without this discriminator, the upsampling generator could learn to ignore the low resolution input video. The conditional generator is trained jointly with the spatial, temporal and matching discriminators. We now describe our conditional generator and matching discriminator.

**Conditional Generator**    The conditional generator takes as input a lower resolution video $\hat{\mathbf{x}}_{\mathbf{w}}^l$, a noise vector $\mathbf{z}$ and optionally a class label $y$, and generates $\hat{\mathbf{x}}_{\mathbf{w}}$, an upsampled version of $\hat{\mathbf{x}}_{\mathbf{w}}^l$. Before feeding it to the generator, we increase the duration of the low resolution video $\hat{\mathbf{x}}_{\mathbf{w}}^l$ to the same temporal duration as the generator output by repeating its frames. Our conditional generator (see Figure 1) stacks units composed by one 3D-ResNet block and two 2D-ResNet blocks. We observe that given enough capacity, the conditional generator with 3D convolutions matches the performance of a similar model trained with ConvGRUs, while 3D convolutions allow for parallel computation over the time axis (see appendix C). Spatial upsampling is performed gradually by progressively increasing the resolution of the generator blocks. To condition the generator we add residual connections (He et al., 2016; Srivastava et al., 2015b) from the low-resolution video to the output of each generator unit. In particular, we sum nearest-neighbor interpolations of the lower resolution input to each unit output. We do not use skip connections for units whose outputs have higher spatial resolution than the lower dimensional video input, i.e. we do not upscale the low resolution video.

**Matching Discriminator (MD)**    MD uses an architecture similar to that of the DVD-GAN temporal discriminator. It discriminates real or generated $(\mathbf{x}_{\mathbf{w}}, \mathbf{x}_{\mathbf{w}}^l)$ pairs. $\mathbf{x}_{\mathbf{w}}$ is downsampled to the same dimensionality as $\mathbf{x}_{\mathbf{w}}^l$, and both inputs are concatenated on the channel dimension. The matching discriminator ensures that the model generates valid upsampled versions of its inputs, and therefore grounds upscaling stages to the low resolution input.

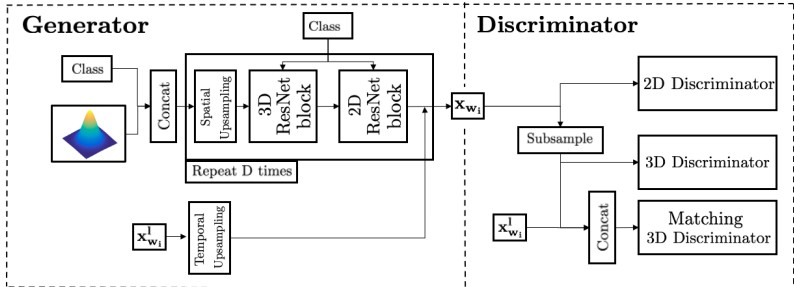

Figure 1: **Stage 2 parametrization** For upsampling stages, we adopt the DVD-GAN (Clark et al., 2019) base architecture and re-purpose it to be conditional on the low-resolution samples. This includes replacing ConvGRUs with 3D convolutions, adding skip connections to $\mathbf{x}_{\mathbf{w_i}}^{\mathbf{l}}$ and adding a matching discriminator that discriminates real or generated ($\mathbf{x}_{\mathbf{w_i}}^{\mathbf{l}}, \mathbf{x}_{\mathbf{w_i}}$) pairs.

## 5  EXPERIMENTS

We consider the Kinetics-600 dataset (Kay et al., 2017; Carreira et al., 2018) to compare our approach with prior art (Clark et al., 2019) in class conditional video generation. Kinetics-600 is a large scale dataset of around 500k Youtube videos depicting 600 action classes. The videos are captured in the wild and exhibit lots of variability. We follow the same data preprocessing of Clark et al. (2019) and use a version of the dataset collected on June 2018. For unconditional video generation, we use the BDD100K dataset (Yu et al., 2018). BDD100K contains 100k videos recorded from inside cars representing more than 1000 hours of driving under different conditions. We use the training set split of 70K videos to train our models.

Defining proper evaluation metrics for video generation is an open research question. We use metrics inspired by the image generation literature and adapted to video. On Kinetics, we report three metrics for comparison to previous works (Clark et al., 2019; Unterthiner et al., 2018): i) Inception Score (IS) given by an Inflated 3D Convnet (Carreira & Zisserman, 2017) network trained on Kinetics-400, ii) Frechet Inception Distance on logits from the same I3D network, also known as Frechet Video Distance (FVD) (Unterthiner et al., 2018), and iii) Frechet Inception Distance on the last layer activations of a I3D network trained on Kinetics-600 (FID). On BDD100K we report the FVD and FID metrics as described before. We do not report IS since there are no well-defined classes in BDD.

Each stage of our model follows the DVD-GAN (Clark et al., 2019) architecture at a given frame resolution with the modifications described in Section 4 for the upsampling stages. We also use the same hyperparameters unless otherwise specified. For the rest of the section we denote video dimensions by their output resolution DxD and number of frames F in the format DxD/F. We use up to 128 nVidia GPUs to train our models. All models are trained with a batch size of 512. Further details can be found in the Appendix.

### 5.1  TWO-STAGE SSW-GAN

In this section we evaluate a two-stage SSW-GAN to empirically validate the proposed training approach. For all experiments we train the first SSW-GAN stage to generate 32x32/25 videos, with a temporal subsampling of 8 frames. The second stage upsamples by a factor of 2 the temporal resolution and by a factor of 4 the spatial resolution, resulting in 128x128/50 videos. During training the second stage operates locally on input windows of 3 or 6 frames and is therefore trained to generate 128x128/6 or 128x128/12 videos, respectively. At inference time we run the model convolutionally over the first stage 25 frames output to generate 128x128 videos of up to 50 frames.

We compare SSW-GAN with DVD-GAN as a baseline on Kinetics-600. We perform this comparison for two DVD-GAN models trained on 128x128/12 and 128x128/48 in Table 1. Additionally, we also ablate our model to evaluate its performance without the matching discriminator in Table 2. All models are trained for 300K iterations on Kinetics (100K for model used in the ablation experiments) and 35k iterations on BDD100K, unless otherwise specified. Samples for our model are shown in Fig. 2 and in the Appendix.

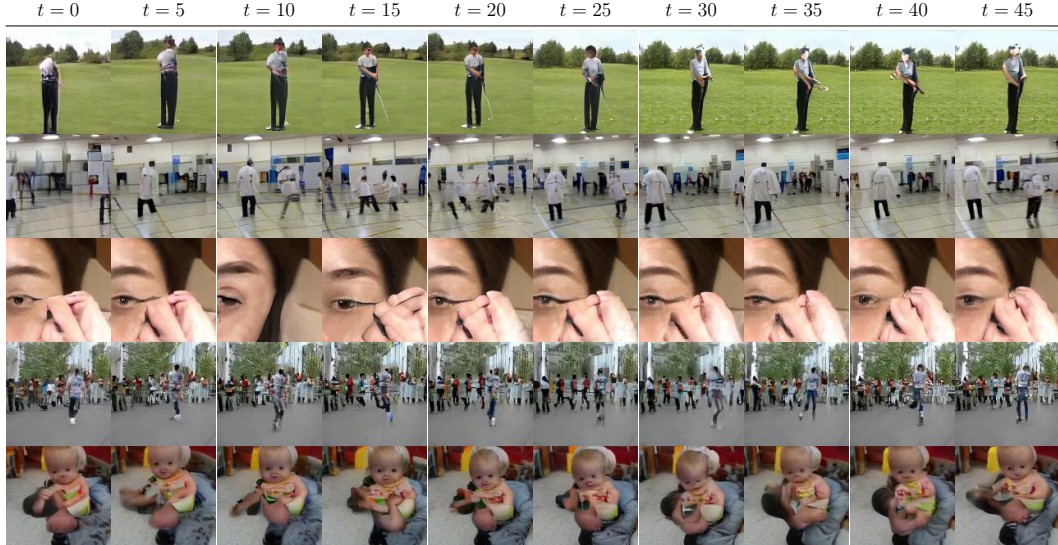

Figure 2: **Randomly selected SSW-GAN 128x128/50 frame samples for Kinetics-600:** We show random samples from our Kinetics-600 128x128/12 model unrolled to generate 50 frames. Each row shows frames from the same sample at different timesteps. We observe that the generations remain fairly consistent through time while the frame quality does not degrade.

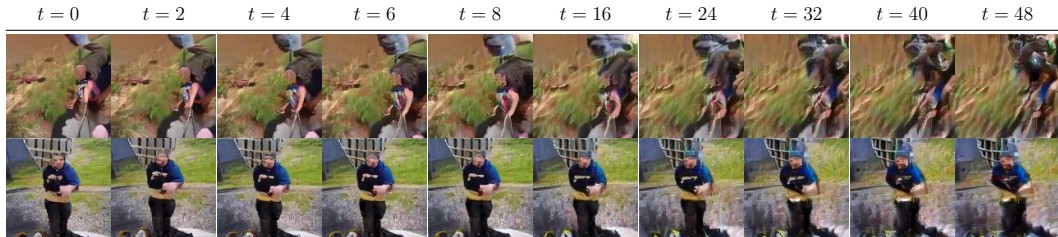

Figure 3: **Unrolling DVD-GAN** We show samples from a 128x128/6 DVD-GAN baseline (our re-implementation) trained on Kinetics-600 and unrolled to generate 128x128/50 videos. While samples are valid for the first 6 frames, they become motionless and degrade thereafter.

**Comparison with prior work** We compare our model trained to generate 128x128/12 videos [1] to the reported metrics for DVD-GAN in Clark et al. (2019) for Kinetics-600 128x128 models. Table 1 shows that our model has better IS score than its DVD-GAN equivalent on 12 frame generations. On 48 frames outputs, SSW-GAN outperforms a 128x128/48 DVD-GAN model in FID score and reaches similar IS score. However, our second stage model is only trained on 128x128/12 outputs, as it is unrolled and applied convolutionally over the first stage output to generate 48 frames. In contrast, DVD-GAN models do not unroll well and tend to produce samples that become motionless past its training horizon (see Fig. 3). SSW-GAN therefore matches DVD-GAN on 48-frame generations in IS on Kinetics-600 and outperforms it in FID, while requiring significantly less computational resources.

**Impact of the temporal window size for the second stage** One of the modelling choices in SSW-GAN is the temporal window length used in the upsampling stages. We compare the 128x128/12 model used in the previous section to a 128x128/6 version of our model. The 128x128/12 model requires approximately twice the amount of GPU memory needed by the 6 frames model, but we expect it to perform better due to the larger input window. As can be seen in Table 1, the 12 frames

---

[1]Stage 1 is trained on 32x32/25 videos. Stage 2 is trained on randomly selected 32x32/6 windows of stage 1 to generate 128x128/12 outputs. During evaluation, the scores obtained for 128x128/12 are obtained by upsampling 6 randomly selected frames from the stage 1. Stage 2 can be unrolled on stage 1 frames to generate 128x128/48 outputs.

Table 1: **Results on Kinetics-600** We compare different versions of our two-stage model against the reported metrics for DVD-GAN models (Clark et al., 2019). Our model when unrolled is able to generate 48 frames of similar quality as a 128x128 DVD-GAN model trained to generate 48 frames while having computational requirements close to the 128x128/12 version. Furthermore, we compare two versions of our model with the second stage trained on different window sizes. We verify that the model with the bigger window size performs better at the cost of being more computationally expensive.

| | 12 frames | | | 48 frames | | |
| Model | IS (↑) | FID (↓) | FVD (↓) | IS (↑) | FID (↓) | FVD (↓) |
|---|---|---|---|---|---|---|
| DVD-GAN 12 frames | 77.45 | **1.16** | - | - | - | - |
| DVD-GAN 48 frames | - | - | - | **81.41** | 28.44 | - |
| Ours - Stage 2 trained on 6 frames | - | - | - | 58.21 | 31.59 | 714.74 |
| Ours - Stage 2 trained on 12 frames | **104.00** | 2.09 | 591.90 | 77.36 | **14.00** | **517.21** |

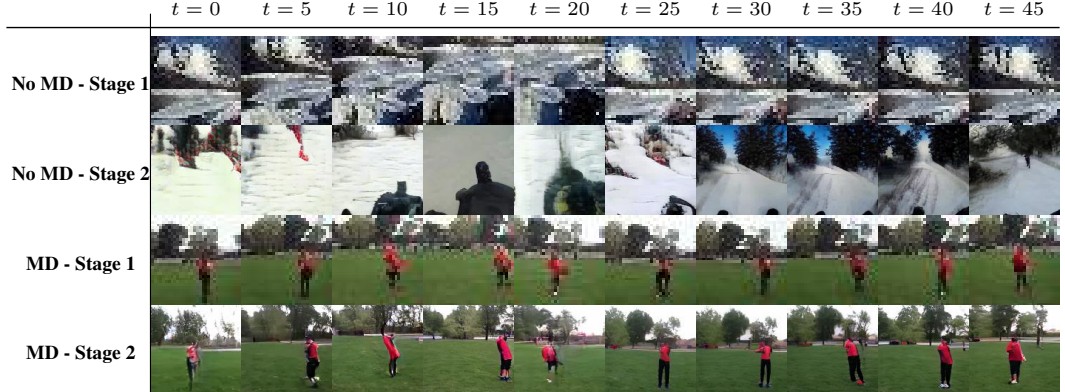

Figure 4: **Matching discriminator** We show here a random sample from our two-stage model on Kinetics with (MD) and without the matching discriminator (no MD). For each sample we show the output of the first stage and the corresponding second stage output. We observe that, while the no MD model generates plausible local snippets at stage 2, it does not remain coherent. Our model generates a coherent sample because it is grounded in the low resolution input.

model obtains significantly higher scores when unrolled to generate 48 frames. We conclude that the window size defines a trade-off between computational resources, since shorter temporal outputs require less computation, and sample quality, as using a short length reduces the temporal field of view of stage 2 and the local independence assumption becomes less accurate.

**Matching discriminator ablation** We now investigate the importance of the matching discriminator in Table 2. On 6 frame generations, SSW-GAN and SSW-GAN (No MD) obtain similar scores. We then unroll both models to generate 50 frames, and observe that SSW-GAN (No MD) generates valid local snippets but is inconsistent through time. Fig. 4 shows an example of the unrolled no MD generations in which this effect is clearly observable. In contrast, our model with a matching discriminator stays consistent through time. This is reflected in the metrics as well. Both models perform similarly at 6 frames, as the model without the matching discriminator is able to generate valid 128x128/6 videos. However, when unrolled this model performs significantly worse than our regular model, as the metrics are capturing the temporal inconsistencies. This justifies the use of a matching discriminator, as it ensures that upsampling stage outputs are grounded to its inputs.

## 5.2 THREE-STAGE SSW-GAN

To show that our model scales with the video dimensionality, we train a three-stage model on the BDD dataset to generate 128x128/100 videos. We train the first stage to output 25 frames at 32x32 resolution with a temporal subsampling of 8 frames. The second stage upscales windows of 12 frames at 64x64 resolution with temporal subsampling of 4 frames (since we are doubling the framerate

Table 2: **Stage 2 128x128/6 comparison** We report metrics on Kinetics and BDD for our model with/out the matching discriminator (MD). Both models perform similarly for 6 frames, corresponding to the training video length. However, adding the MD improves the 50-frame generations, as they remain grounded to the first stage output allowing for consistent videos beyond the training length.

| Dataset | Model | 6 Frames | | | 50 Frames | | |
|---|---|---|---|---|---|---|---|
| | | IS (↑) | FID (↓) | FVD (↓) | IS (↑) | FID (↓) | FVD (↓) |
| Kinetics-600 | DVD-GAN (our reimpl.) | 41.2 | 1.60 | 841.5 | N/A | N/A | N/A |
| | SSW-GAN (No MD) | **50.31** | 1.62 | 594.99 | 37.81 | 42.29 | 1037.79 |
| | SSW-GAN | 48.44 | **1.06** | **565.95** | 49.44 | 31.87 | 790.97 |
| BDD100K | SSW-GAN (No MD) | - | 1.36 | 211.69 | - | 26.52 | 575.51 |
| | SSW-GAN | - | **1.07** | **144.96** | - | **18.73** | **326.78** |

Table 3: **Computational costs:** We compare the required memory and training time for a a DVD-GAN model and its SSW-GAN equivalent. Our model takes roughly half of the training time and requires 4x less GPU memory. Data for DVD-GAN is based in our reimplementation of the model.

| | Samples/GPU | Memory (MB) | Iteration time (s) | Training time |
|---|---|---|---|---|
| Stage 1 32x32/25 | 4 | 27897 | 3.45 | 12 days |
| Stage 2 128x128/12 | 4 | 29073 | 3.2 | 11 days |
| DVD-GAN 128x128/50 | 1 | 32203 | 2.975*4 | 41 days |

of the first stage). The third stage is trained to upscale 12 frame windows at 128x128 resolution for a final temporal subsampling of 2 frames. A full pass for a single example in a DVD-GAN 128x128/100 model would require more than 32GB of memory, beyond the limits of most commercial GPUs. In contrast, we are able to train our model using batch size 512 with 128 GPUs. Samples from this model can be seen in Figure 5.

## 6 CONCLUSIONS

We propose SSW-GAN, a multi-stage video generator that outputs long high-resolution videos, while being more computationally efficient than equivalent single-stage GAN approaches. High capacity models trained with big amounts of data are key aspects for video generation. SSW-GAN is a step towards scalable models and allows training SOTA video GANs with fewer resources.

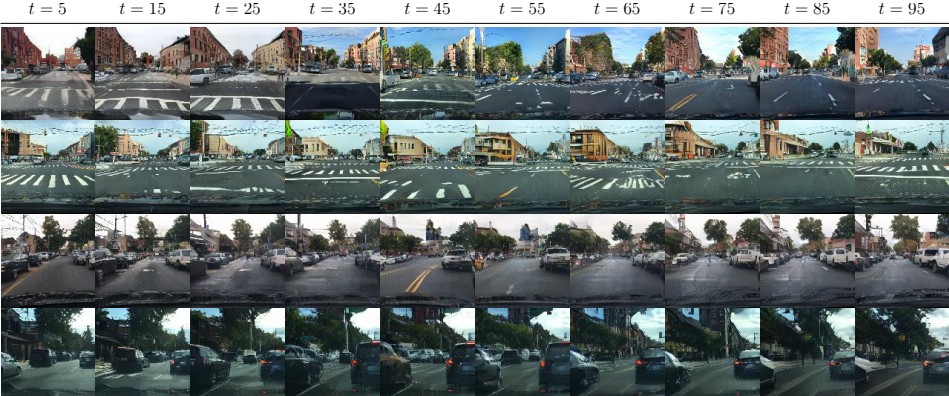

Figure 5: **Random 128x128/100 BDD100K samples:** We show samples from our three-stage BDD100K model. Each row shows a different sample over time. Despite the two stages of local upsampling, the frame quality does not degrade noticeably through time.

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

# A   ADDITIONAL IMPLEMENTATION DETAILS

We use the Adam optimizer Kingma & Ba (2014) with learning rate $\lambda_G = 1e10^{-4}$ and $\lambda_D = 5e10^{-4}$ for the generator and the discriminator, respectively. The discriminator is updated twice for each generator update.

We use orthogonal initialization for all the weights in our model and use spectral normalization both in the generator and the discriminator. We only use the first singular value to normalize the weights. Different from DVD-GAN, we do not use weight moving averages nor orthogonal penalities.

Conditional batch normalization layers use the input noise as the condition, concatenated with the class label when applicable. Features are normalized with a per-frame mean and standard deviation.

To unroll a generator beyond its training temporal horizon, we apply it convolutionally over longer input sequences. We perform 200 "dummy" forward passes to recompute the per-timestep batch normalization statistics at test time.

All convolutions in our models use 3x3 or 3x3x3 filters with padding=1 and stride=1, for 2D or 3D convolutions respectively. All models were implemented in PyTorch.

# B   MODEL ARCHITECTURE DETAILS

For the rest of the section, we use B to denote the batch size, T for the number of frames or timesteps, C for the number of channels, H for the height of the frame and W is the width of the frame.

## B.1   STAGE 1 ARCHITECTURE

Our first stage model is based on a re-implementation of DVD-GAN Clark et al. (2019). In all our experiments, the first stage produces 32x32/25 outputs.

**Generator**   The generator is composed by a stack of units where each unit is comprised of a ConvGRU layer and two 2D-ResNet upsampling blocks. We follow the nomenclature of Brock et al. (2018); Clark et al. (2019) and describe our network using a base number of channels $ch$ and the channel multipliers associated with each unit. Our stage-1 generators is formed by 4 units with channel multipliers $[8, 8, 4, 2]$. The base number of channel is $128$.

The first input of this network is of size BxTx(8x$ch$)x4x4. This input is obtained by first embedding the class label onto a 128 dimensional space, then concatenating the embedding to a 128 dimensional noise vector. This concatenation is mapped to a Bx(8x$ch$)x4x4 tensor with a linear layer and a reshape, and then the final tensor is obtained by replicating the output of the linear layer T times.

The ConvGRU layer Ballas et al. (2015) follows the ConvGRU implementation of Clark et al. (2019) and uses a ReLU non-linearity to compute the ConvGRU update.

The 2D ResNet blocks are of the norm-act-conv-norm-act-conv style. We use conditional batch normalization layers, ReLU activations and standard 2D convolutions. Before the first convolution operation and after the first normalization and activation, there is an optional upsampling operation when increasing the resolution of the tensor. We use standard nearest neighbor upsampling. Except for the last unit, all units perform this upsampling operation. The conditional batch normalization layers receive the embedded class label (if applicable) and the input noise as a condition and map it to the corresponding gain and bias term of the normalization layer using a learned linear transformation. The 2D ResNet blocks process all frames independently by reshaping their input to be (B*T)xCxHxW.

The output of the last stack goes through a final norm-relu-conv-tanh block that maps the output tensor to RGB space with values in the [-1, 1] range.

**Discriminator**   There are two discriminators, a 2D discriminator and a 3D discriminator. The 2D discriminator is composed of 2D ResNet blocks. Each ResNet block is formed by a sequence of relu-conv-relu-conv layers. There are no normalization layers in the discriminator. After the last conv in each block there is an optional downsampling operation, which is implemented with average

pooling layers. The 2D discriminator receives as input 8 randomly sampled frames from real or generated samples.

The 3D discriminator is equal to the 2D discriminator except that its first two layers are 3D ResNet blocks, implemented by replacing 2D convolutions with regular 3D convolutions. The 3D discriminator receives as input a spatially downsampled (by a factor of two) real or generated sample. The 2D blocks process different timesteps independently.

We concatenate the output of both discriminators and use a geometric hinge loss. The loss is averaged over samples and outputs.

We use $128$ as base number of channel for both discriminators, with the following channel multipliers for each ResNet block: $[16, 16, 8, 4, 2]$

## B.2 Upsampling stage architecture

The upsampling stage models follow the same architecture as the first stage with the following modifications.

**Generator** The generator units replace the ConvGRU layers with a Separable 3D convolution. We first convolve over the temporal dimension with a 1D temporal kernel of size 3 and then convolve over the spatial dimension with 2D 3x3 kernel. We empirically compare generators with ConvGRU and separable convolutions in section C.

We add residual connections at the end of each 3D and 2D ResNet block to an appropriately resized version of $\mathbf{x}^l_{w_i}$. We use nearest neighbor spatial downsampling for this operation, and we use nearest neighbor temporal interpolation to increase the number of frames of $\mathbf{x}^l_{w_i}$. We then map the residual to the appropriate number of channels using a linear 1x1 convolution. We do not add $\mathbf{x}^l_{w_i}$ residual connections to feature maps with spatial resolutions (HxW) greater than the resolution of $\mathbf{x}^l_{w_i}$.

**Discriminator** We reuse the same 2D and 3D discriminators as for the first stage. Additionally, we add a matching discriminator that discriminates $(\mathbf{x}^l_{w_i}, \mathbf{x}_{w_i})$ pairs. The matching discriminator utilizes the same architecture as the 3D discriminator. It receives as input a concatenation of $\mathbf{x}^l_{w_i}$ and a downsampled version of $\mathbf{x}_{w_i}$ to match the resolution of $\mathbf{x}^l_{w_i}$. We concatenate the outputs of all three networks and use a geometric hinge loss as for the first stage discriminator. The overall loss is averaged over samples and output locations.

For 128x128 generations on Kinetics, the generator uses $128$ as base number of filters with the following channel multipliers $[8, 8, 4, 2, 1]$. All discriminators have 96 base channels and the following channel multipliers $[1, 2, 4, 8, 16, 16]$. All our models at 128x128 are two-stage models. We train models to upsample $\mathbf{x}^l_{w_i}$ of sizes 32x32/3 or 32x32/6 to 128x128/6 or 128x128/12, respectively. Since we train our first stage for 32x32/25 outputs, our two-stage models can generate 128x128/50 outputs when unrolled.

For 128x128 generations on BDD100K, the generator uses $96$ as the base number of channels with channel multipliers $[8, 4, 4, 2, 2]$. All discriminators have $64$ base channels and channel multipliers $[1, 2, 4, 8, 8, 16]$. Our 128x128 models upsample 64x64/6 $\mathbf{x}^l_{w_i}$ inputs to 128x128/12, and can generate outputs of up to 128x128/100.

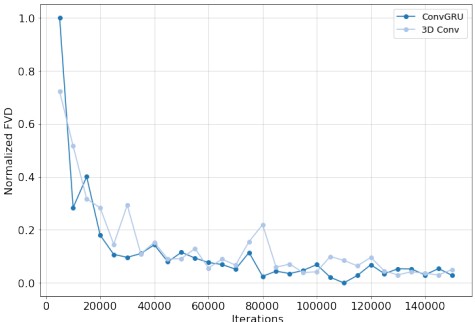 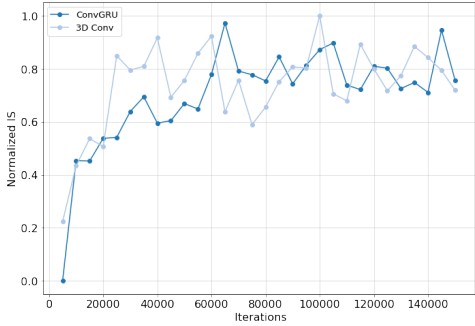

Figure 6: **Comparison of recurrent layers** We compare two variants of the same generator, one with a single ConvGRU layer per generator block and one with a separable 3D convolution per generator block. On the left we show the evolution of the FVD score during training, and on the right we show the Inception Score. Both scores are normalized to the [0, 1] range where 1 is the highest score obtained by these models and 0 the lowest. Both models have similar behaviour and computational costs, but the 3D convolution processes inputs in parallel.

## C    COMPARISON OF RECURRENT LAYERS

In this section we justify the change of the ConvGRU for Separable 3D convolutions in upsampling stages. In Figure 6 we compare the evolution of two metrics (IS and FVD) during training for two variants of the same two-stage model, one using ConvGRUs and one using separable 3D convolutions. Both models show similar behavior during training and achieve similar final metrics. However, ConvGRUs perform sequential operations over time whereas 3D convolutions can be parallelized.

## D    ADDITIONAL SAMPLES

Additional samples can be found in .mp4 format along with this appendix in the supplementary materials file. These videos show multiple samples from our two-stage model for Kinetics and our three-stage model for BDD. For each sample, we show the output of each stage in a row.

We have included 3 videos. One video shows samples from our 128x128/12 model unrolled to generate 128x128/100 samples on BDD. Another video shows samples from our 128x128/12 model unrolled to generate 128x128/50 samples on Kinetics. Finally, we include a video with samples from the same model but without the matching discriminator (baseline no matching disc suffix filename).

We include some additional samples for our Kinetics 128x128/12 model and BDD 128x128/12 model below.

For all evaluations, we sample from an isotropic Gaussian with unit variance for ease of comparison and reproducibility. Samples for figures and the provided video files were produced by sampling with standard deviation $\sigma = 0.5$. We observed that noise samples with reduced variance produce higher quality samples but are slightly less diverse.

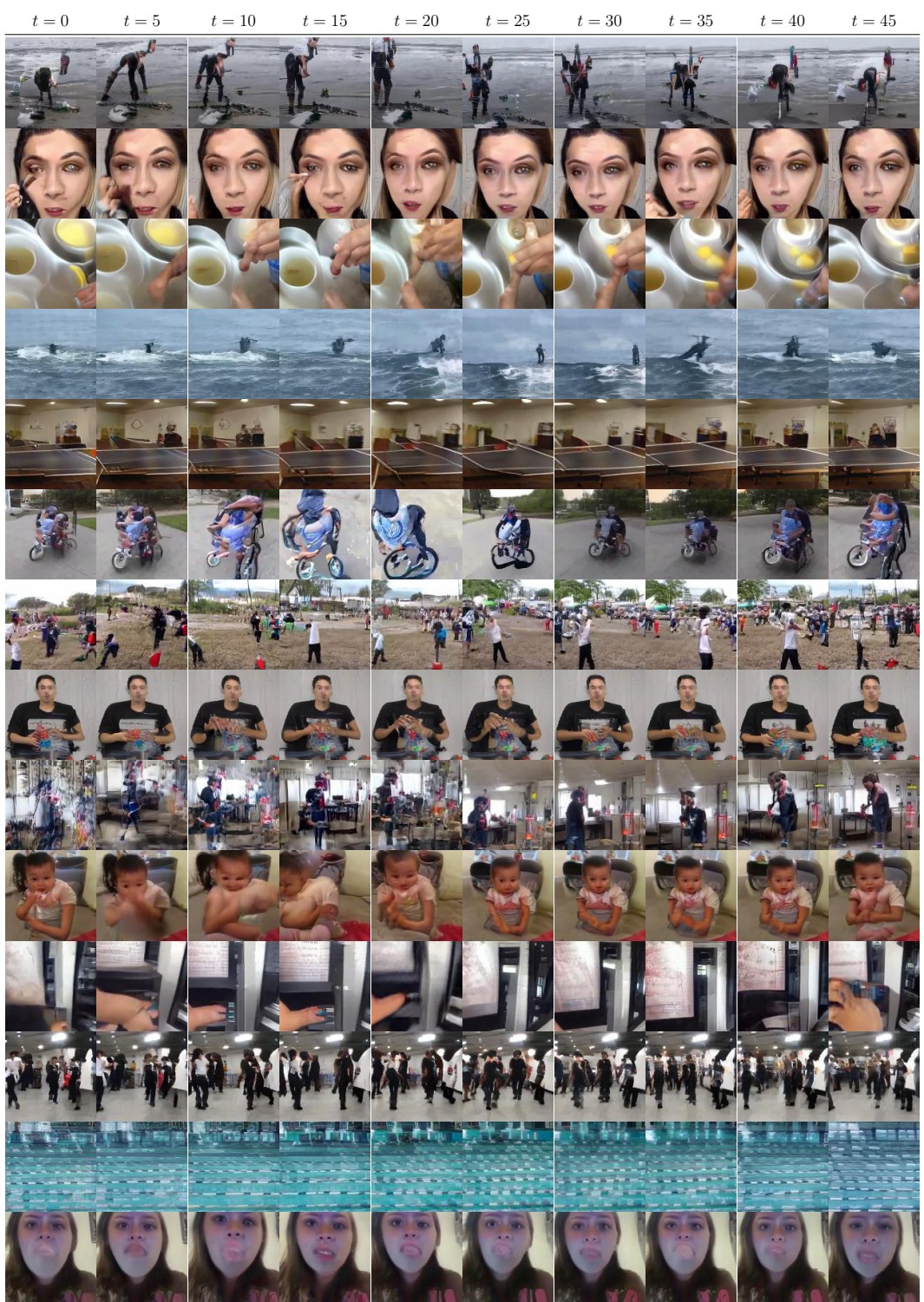

Figure 7: **Additional samples for Kinetics 128x128/12** We show additional samples from our two-stage Kinetics 128x128/12 model unrolled to generate 128x128/50 videos. More samples can be found in the supplementary videos.

$t = 5$ $t = 15$ $t = 25$ $t = 35$ $t = 45$ $t = 55$ $t = 65$ $t = 75$ $t = 85$ $t = 95$

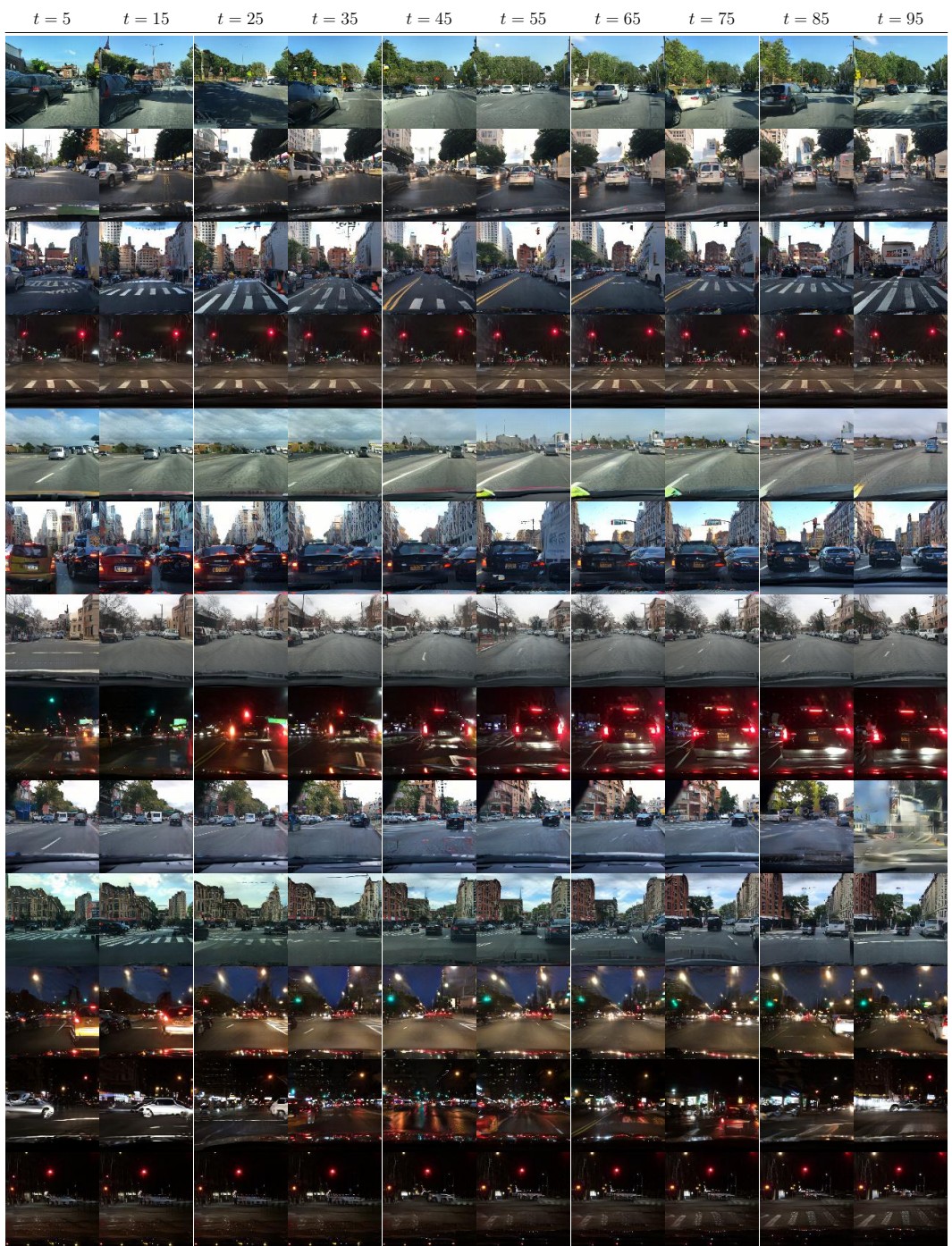

Figure 8: **Additional samples for BDD 128x128/100** We show additional samples from our three-stage BDD 128x128/12 model unrolled to generate 128x128/100 videos.

# E  POWER SPECTRUM DENSITY

Some video generation models tend to produce blurry results over time. In this section we generate Power Spectrum Density (PSD) plots, following (Ayzel et al., 2020) to assess whether our generations become blurrier over time.

We conduct this experiment on Kinetics for videos of 50 frames at 128x128 pixels resolution. We use our model with the first stage trained on 32x32/25 sequences and the second stage trained to generate 128x128/12 video snippets from 32x32/6 windows, and unrolled after training to produce 128x128/50 videos. We took 1800 random videos from the ground truth data (GT) and 1800 generations from our model. We compute the PSD at frames 1, 10, 25, and 50 of each video. For each set of 600 videos, we compute the average PSD across videos, on a per frame basis. Finally, we use the three sets of 600 videos to compute the standard deviation and mean for the average PSD of the original data and our generations. Figure 9 shows the plots for different frame indices. We observe that our generations have a very similar PSD to that of the ground-truth data in all video frames. This indicates that our generations, while they might not be accurate, have very similar frequency statistics as the ground-truth data. We do not observe any significant blurring over time, and this is reflected in the plots, showing that even for frame 50 the high frequency part of the PSD is very similar between the original data and our generations.

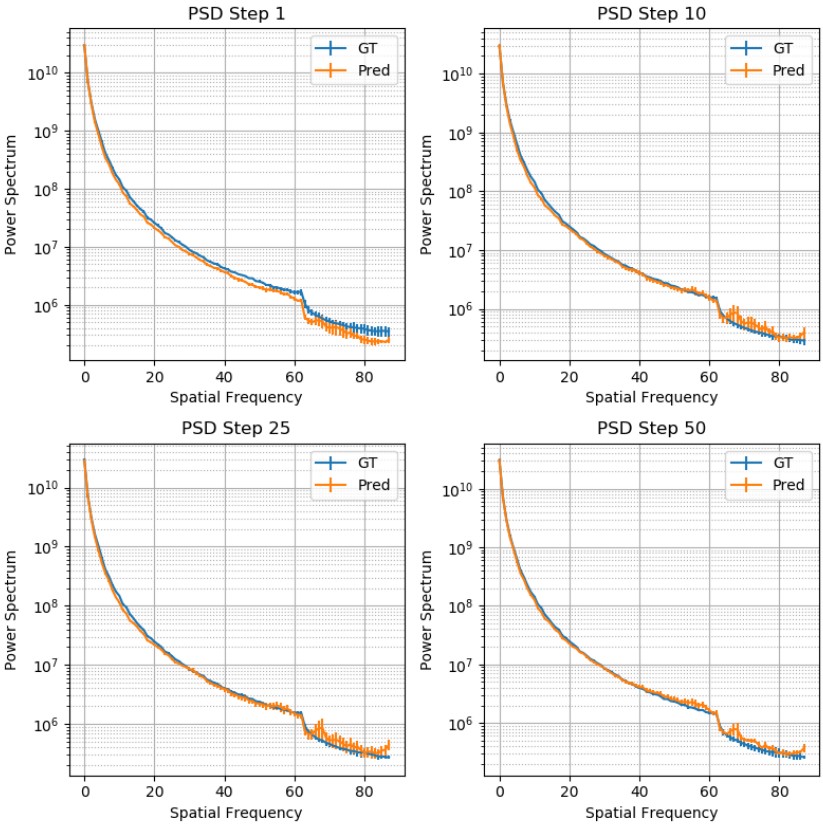

Figure 9: **Power Spectrum Density (PSD) plots for different time steps** We show a comparison of the PSD between the original data and our generations at different steps in the predictions. We observe that our generations have a similar PSD to that of the original data, even at the end of the generation, indicating that the generations do not blur over time significantly.

## F UPSAMPLING VISUALIZATIONS

Here we show some examples of stage 1 generations on Kinetics-600 for a 32x32/25 model, as well as the corresponding 128x128/50 generations from a stage 2 trained to upscale 32x32/6 windows to 128x128/12 and unrolled over the whole first stage generation. Examples are shown in Figure 10, in which, for each example, we show the stage 1 generation on the top row and the corresponding stage 2 generation in the lower row. We observe that the second stage adds details and refines the low resolution generation, but at the same time keeps the overall structure of the low resolution generation and is properly grounded.

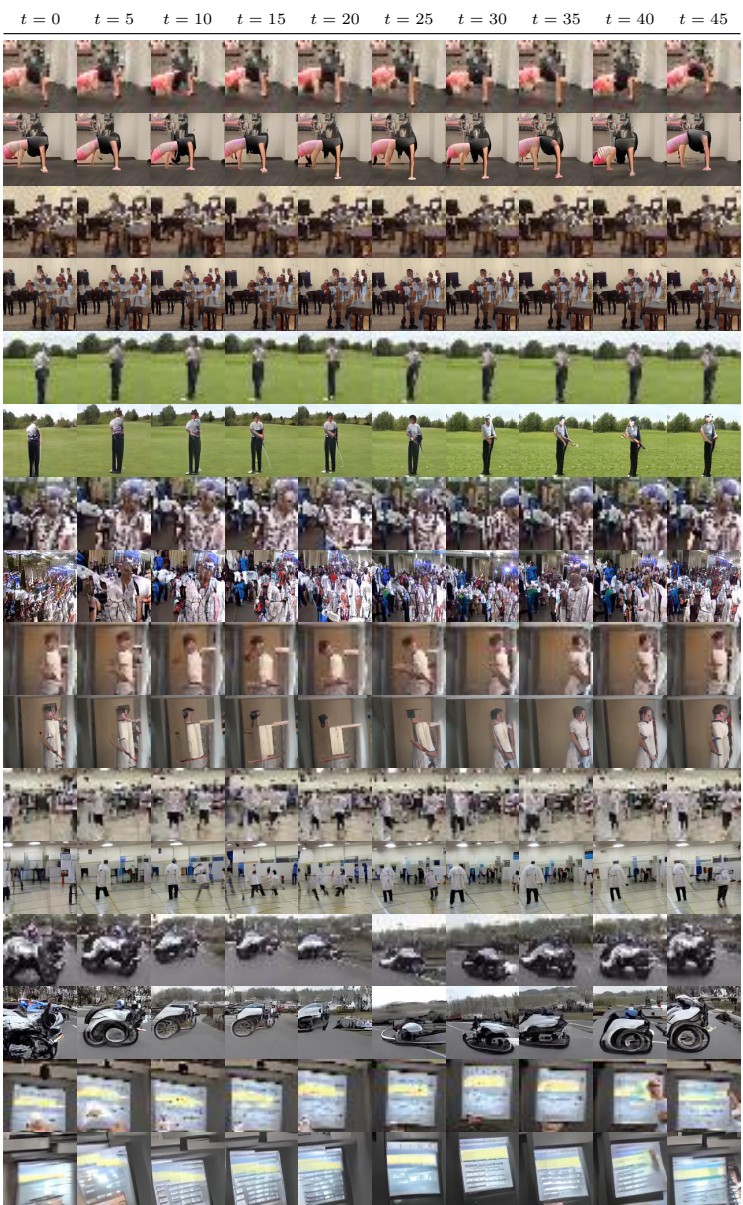

Figure 10: **Pairs of samples from stage 1 and their corresponding stage 2 output** We show a few examples from our 128x128/12 two-stage model trained on Kinetics-600 and unrolled to generate 128x128/50 videos. For each example, we show the first stage low resolution generation and the corresponding stage 2 upsampling. Stage 2 outputs refine the details of the first stage generations but retain the overall scene structure.

## G    INFLUENCE OF MOTION ON THE RESULTS

In this section we analyze whether our model has different performance for categories with different motion characteristics.

We conduct this experiment for the our Kinetics-600 model trained to generate 32x32/25 videos in the first stage to upscale 32x32/6 videos to 128x128/12 videos for the second stage, which we then subsequently unroll over the full first stage generation to obtain 128x128/50 videos.

We randomly selected five categories of videos with high motion content (bungee jumping, capoeira, cheerleading, kitesurfing and skydiving) and five categories with less dynamic videos (doing nails, cooking egg, crying, reading book and yawning). We then generated 1000 samples from our model for each category, and used all the available samples in the dataset to compute IS and FID scores per category.

Table 4 shows the IS and FID scores of each class, while Figure 11 and Figure 12 show samples from high motion and low motion categories, respectively. We do not observe a significant trend that indicates that the model produces worse generations for high motion classes. Some low motion categories have high FID scores similar to the highest scores for the high motion categories, while on average the IS scores for the high motion categories are slightly better. We also do not notice a qualitative difference in the samples. Instead, there might be other factors (such as the amount of structure present in a scene) that have a greater impact on the quality of the generations.

Table 4: **Per category scores for classes with different amounts of motion (Kinetics-600)** We report per-class IS and FID scores for 5 randomly selected categories with high motion and 5 categories with low motion. We observe that there is a high variability in FID scores, with some classes with low motion having high scores as well as some high motion classes. In IS scores there are few differences between the two groups, with the high motion group having a slightly higher mean score.

|  | class | is (↑) | fid (↓) | # Videos |
|---|---|---|---|---|
| **High Motion** | Bungee Jumping | 11.66 | 82.21 | 799 |
|  | Capoeira | 9.48 | 84.57 | 816 |
|  | Cheerleading | 12.10 | 116.84 | 982 |
|  | Kitesurfing | 11.36 | 108.81 | 648 |
|  | Skydiving | 5.99 | 90.25 | 983 |
| **Low Motion** | Doing Nails | 13.32 | 91.67 | 537 |
|  | Cooking Egg | 7.60 | 111.63 | 441 |
|  | Crying | 8.92 | 70.74 | 627 |
|  | Reading Book | 11.13 | 64.97 | 793 |
|  | Yawning | 9.71. | 79.08 | 530 |

$t = 0$ $\quad$ $t = 5$ $\quad$ $t = 10$ $\quad$ $t = 15$ $\quad$ $t = 20$ $\quad$ $t = 25$ $\quad$ $t = 30$ $\quad$ $t = 35$ $\quad$ $t = 40$ $\quad$ $t = 45$

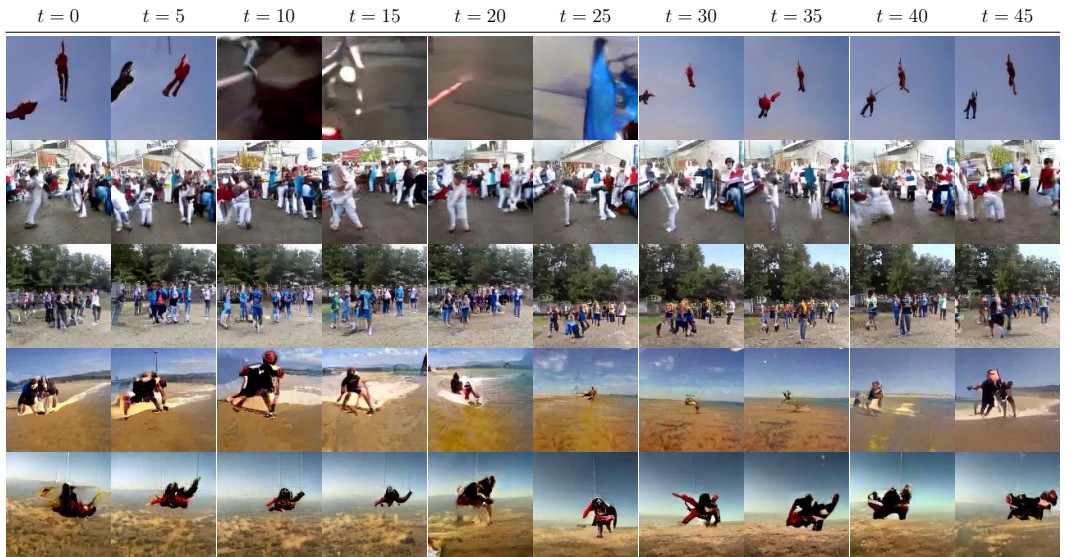

Figure 11: **Samples from Kinetics-600 classes with high motion content**

$t = 0$ $\quad$ $t = 5$ $\quad$ $t = 10$ $\quad$ $t = 15$ $\quad$ $t = 20$ $\quad$ $t = 25$ $\quad$ $t = 30$ $\quad$ $t = 35$ $\quad$ $t = 40$ $\quad$ $t = 45$

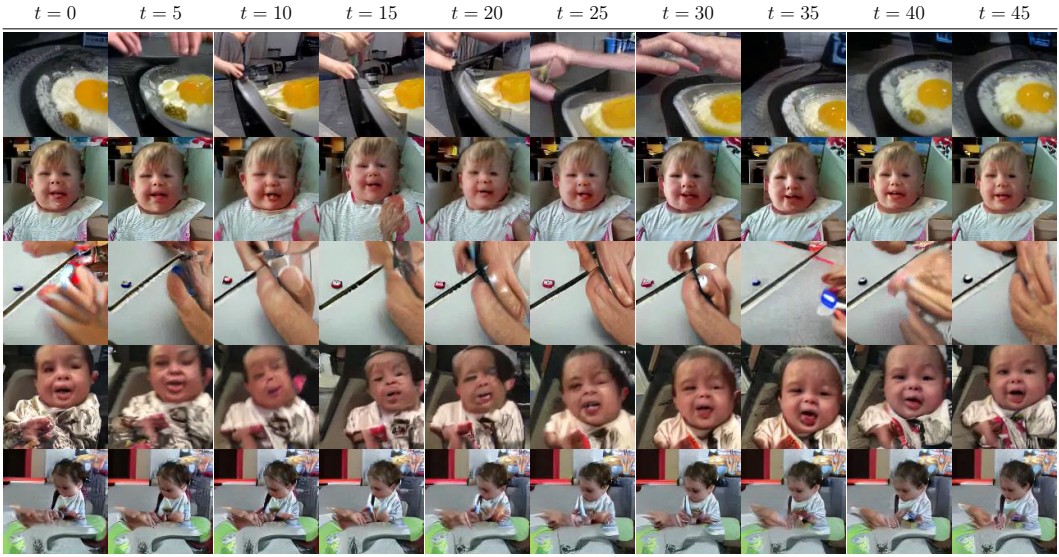

Figure 12: **Samples from Kinetics-600 classes with low motion content**

## H    JOINT TRAINING OF MULTIPLE STAGES

Our proposed method consists of multiple stages trained independently and sequentially. This allows us to reduce the memory requirements, as we do not have to keep the intermediate activations to backpropagate through the full model, and allows us to define stages with larger memory requirements than feasible if the model was trained end-to-end. However, our approach does not require each stage to be trained independently. Furthermore, with end-to-end training we could fit an equivalent full model in a single optimization round, thus reducing the training time, and potentially finding a better solution. Additionally, we might not need to discriminate the output of intermediate stages nor use matching discriminators. In this section we report some initial results investigating the joint training of the different stages.

We use a two-stage model trained on Kinetics-600. The first stage is trained to generate 32x32/6 videos, and the second stage is trained to upscale 32x32/3 snippets to 64x64/6 videos. When unrolled over the full first stage generation, this model is capable of producing 64x64/12 videos. We only discriminate on the output of the second stage, with a spatial and temporal discriminator. We don't use any matching discriminator in this experiment.

Figure 13 shows some generations from this model. We observe that the output of the first stage is no longer a valid low resolution video. However, the second stage learns to generate valid video snippets. In general we observe that the generations from this model do not exhibit lot of motion. This could be due to the window size used in the experiment or to the joint training setup.

A potential avenue to improve joint training would be to use discriminators on the first stage output and/or use a matching discriminator to discriminate pairs of first and second stage outputs.

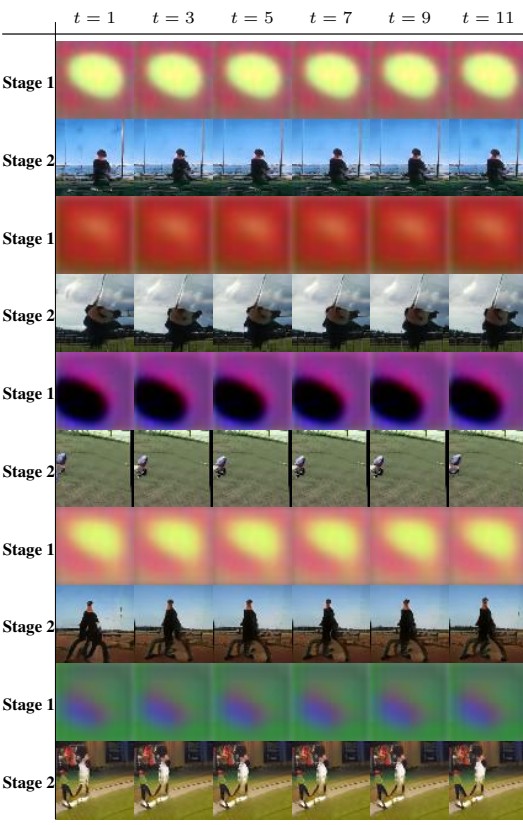

Figure 13: **64x64/12 samples from the jointly trained model on Kinetics-600** We show unrolled samples from a variant of our model with all stages trained jointly. The output of the first stage is not a valid generation anymore, but the second stage learns to generate valid video snippets. Samples do not exhibit lot of motion.

## I  BROADER IMPACT

Here we discuss the broader ethical impacts of this work. SSW-GAN is a generative model for video. As with other generative models, there is a chance that similar methods as the one we propose might be used to create "deepfakes". Note however that it would require extensive follow-up work and that it would not be a direct application of our methods. Furthermore, it is not possible to know in advance what a generation is going to look like, and therefore our model could produce results that could be nonsensical or in bad taste.

At the same time, generative models for video have multiple positive applications. First, their generations can be used to train better classifiers. They can also be used to facilitate content creation for visual artists. Furthermore, variants of our model could be used to better compress videos, which has advantages such as reduced bandwidth requirements to transmit video data.

Overall our model would require follow-up work to enable some of the positive and negative applications described, as in its current form it is a class-conditional generative model with its main ability being that of generating samples similar to those used for its training.

