# OpenReview forum: "SSW-GAN: Scalable Stage-wise Training of Video GANs"
_ICLR.cc/2021/Conference — Reject_

### Official Review · AnonReviewer4 · 2020-10-26
**Generated samples not temporally consistent**

**Rating:** 6
**Confidence:** 5

**Review:**

The paper proposes a GAN-based model which generates videos in multiple stages. The main idea is the upsampling of the spatio-temporal resolution upon addition of a stage. This is the key feature of the proposed model allowing the model to generate videos of higher temporal resolution while using significantly less computational resources.

**Strengths**
+ The paper is clearly written
+ The model performs competitively with relevant baselines with respect to quantitative metrics
+ The evaluation of the model has been conducted on real world datasets
+ Implementation details have been mentioned clearly

**Weaknesses**
- There have been earlier attempts for multi-stage video generation  [1,2]. However, the paper misses citations in this direction. Also, apart from condition for the generation, how is the proposed model different from the existing multi-stage ones?
- The generated samples for Kinetics dataset are not temporally consistent and misses several details especially for smaller entities in video. To list a few: in Figure 2 row 4, the baby's face looks distorted and different in every frame; in Figure 3 row 2 and in Figure 2 row 1, the face of the person is completely incomprehensible.
- The generated samples in the paper do not have a lot of perceived motion in them. How does the model perform when the input class is supposed to possess huge temporal variations?

Overall, the paper presents a scalable way to generate video with higher temporal resolution. However, the generated results do not look realistic and lot of important details are missing in the generated samples. Therefore, my initial rating for this paper is 4.


*References used in the review:*

[1] Learning to generate time-lapse videos using multi-stage dynamic generative adversarial networks. In Proceedings of the IEEE Conference on Computer Vision and Pattern Recognition, 2018

[2] Zhao, L., Peng, X., Tian, Y., Kapadia, M. and Metaxas, D.N., Towards Image-to-Video Translation: A Structure-Aware Approach via Multi-stage Generative Adversarial Networks. International Journal of Computer Vision, 2020



============================================**Post-Rebuttal Comments**==================================

I appreciate the revisions and additional results presented by the authors. The authors have addressed my concerns as well as improved the clarity of the model description in the revised version of the paper. While that results are not perfect, I acknowledge that the problem of video generation is difficult and I believe such multi-stage model can motivate future methods in this direction of scalable video generation. Therefore, I would like to improve my score to 6 and would recommend acceptance of this paper.

---

> ### Author Response · Authors · 2020-11-14
> **Answer to AnonReviewer #4 (part 1)**
>
> We thank you for your effort in reviewing our paper.
>
> **Earlier attempts for multi-stage video generation**
> We will include both citations in the paper. We note however that both references have a different motivation to our work and are significantly different. We summarize them below:
>
> Reference [1] proposes a multi-stage video prediction model. The first stage aims at generating realistic images, while the second stage models dynamics. This is similar in spirit to other approaches in which motion and content are disentangled (for example MoCoGAN or the dual discriminator of DVD-GAN). In contrast, the goal of our multi-stage generator is to have a coarse-to-fine approach to video generation that allows for higher resolution long videos. Our model arguably matches the SOTA in video generation (DVD-GAN) while requiring less resources, and we are able to generate videos for 100 frames which are significantly longer temporal horizons than both [1] (32 frames) and DVD-GAN (up to 48 frames).
>
> Reference [2] considers the problem of image-to-video translation. Their method works by first producing a motion structure, and then the second stage refines that generation to produce frames. Similarly to [1], the goal of this work is not to propose a scalable alternative to training video GANs, and instead their stages focus on modeling different aspects of a video.
>
> In summary, both approaches aim at decomposing the generative process into different semantic stages, while the goal of our model is to provide a scalable alternative to SOTA video GAN models that uses a coarse-to-fine decomposition to achieve this goal.
>
> **Temporal consistency on Kinetics ... generations missed several details**
>
> We would like to point out that video generation on complex datasets is a hard task which is not solved. The quality of video generative models is far from being at the level of their image counterparts.
>
> We agree that the consistency or fine-grained details in some of our generations could be improved. Nevertheless, the quantitative evaluations indicate that the samples generated by our model are of similar or better quality than those from DVD-GAN, as you noticed in your review, and DVD-GAN is the current state-of-art in video generation as far as we know. Our samples also are of similar qualitative quality than DVD-GAN.  For reference, these are some 128x128 generations from DVD-GAN using the truncation trick, provided by the DVD-GAN authors. They also show inconsistencies or lack of details similar to the ones in our results (bottom row, second column for instance): https://drive.google.com/file/d/1P8SsWEGP6tEGPPNPH-iVycOlN6vpIgE8/view.
>
> The goal of our paper is to provide a more scalable alternative to current video generation GANs to allow us to generate longer videos, while maintaining the sample quality.
>
> **Missing motion**
> We randomly selected 5 classes with high motion/temporal variations and 5 classes with low motion and computed per-class scores. For each category we generated 1000 random samples and used all the available videos in that category (usually between 500 to 1000 examples) to compute the scores. We observe high variability in FID and IS across classes, but we do not observe significant trends within high motion or low motion classes. For example, there are categories in the non-motion category with FID score > 110, and the IS scores are on average slightly better for the motion category. Looking at the samples, we also do not observe any significant distinctions between the different groups of classes.
>
> Instead we noticed differences in the generations across all classes in the dataset. We believe the class variability might be due to other factors such as amount of data available, having more semantically related classes or the amount of structure present in a particular class - generating simple objects is easier than generating faces.
>
> We will include this discussion with some visualizations in an updated manuscript. Below we provide the per-class scores we obtained:
>
> **Low Motion group**
>
> Category | FID | IS | # examples
> --------------------------
> Cooking_egg | 111.63 | 7.60 | 441
>
> Crying | 70.74 | 8.92 | 627
>
> Doing_nails | 91.67 | 13.32 | 537
>
> Reading_book | 64.97 | 11.13 | 793
>
> Yawning | 79.08 | 9.71 | 530
>
> **High Motion group**
>
> Category | FID | IS | # examples
> --------------------------
> Bungee_jumping | 82.21 | 11.66 | 799
>
> Capoeira | 84.57 | 9.48 | 816
>
> Cheerleading | 116.84 | 12.10 | 982
>
> Kitesurfing | 108.81 | 11.36 | 648
>
> Skydiving | 90.25 | 5.99 | 983

---

> ### Author Response · Authors · 2020-11-14
> **Answer to AnonReviewer #4 (part 2)**
>
> In summary, we show that our results are in line with current SOTA video GAN models on a hard dataset of videos in the wild such as Kinetics. Current SOTA models have huge computational requirements -  our model matches their performance while requiring 2x less resources and at the same time we are able to generate up to 100 frames at 128x128 pixels, more than 2x the amount of frames that could be generated by DVD-GAN.
>
> We thank you again for your review. We hope that we addressed your concerns. Let us know if you have further questions.

---

### Official Review · AnonReviewer2 · 2020-10-27
**This paper proposes a stage-wise strategy to train Generative Adversarial Networks for videos. The contribution of this paper is very limited and the experiments are not convincing.**

**Rating:** 3
**Confidence:** 4

**Review:**

Pros:
1. A stage-wise approach to train GANs for video is defined to reduce the computational costs needed to generate long high resolution videos.
2. The authors provide some quality results of the proposed approach.

Cons:
1. The contribution of this paper is very limited. The authors just do some incremental improvement based on current GAN models, and the theoretical analysis for the stage-wise training approach is not enough.
2. The experiments are not convincing. The authors only compared the baseline methods in the experiments. Besides, the proposed training strategy should be applied in different generation models based on GAN to show the effectiveness in different cases.
3. This paper aims to reduce the computation cost of the model training, but do not achieve significant effect, which takes 23 days for model training.

---

> ### Author Response · Authors · 2020-11-14
> **Answer to AnonReviewer #2**
>
> Thank you for taking the time to review our paper. We respond to your different concerns below.
>
> 1. To the best of our knowledge, we are the first to propose a stage-wise generative model for video where **each stage is trained independently**. We show that our approach is competitive with DVD-GAN, which is the current state-of-the-art for video generation as far as we know. Furthermore, we demonstrate that our approach can generate videos of up to 100 frames at 128x128 resolution. Our method is the first one to produce generations at such scale. Finally, we show that approach is theoretically grounded.  We believe that these contributions are novel and significant.
> We will happily discuss the novelty in relation with other related work if you could provide us with some references.
>
> 2. We empirically validate our approach on Kinetics-600 and BDD100K, two large-scale datasets with complex videos in real-world scenarios. In addition to baselines based on our approach, we compare to DVD-GAN which is arguably the current state-of-art for video generation. Our approach matches or outperforms state-of-art approaches while requiring significantly less computational resources. We are not aware of a peer-reviewed generative model that outperforms DVD-GAN for video generation.
>
> 3. Even though the computational cost is still important, our approach is nonetheless a 2x improvement over current SOTA models which is significant. It is a step forward in having more tractable video models. Furthermore, thanks to the computational savings, we are able to generate videos for longer temporal horizons (100 frames) than previous work on video generation.
>
> Thank you for taking the time to review our paper. We hope that our answers clarify the significance and value of our contributions. We hope that you will reconsider your score, which we believe is severe with our work. Let us know if you have any further questions or concerns.

---

### Official Review · AnonReviewer1 · 2020-10-27
**Good contribution in improvement of DVD-GAN**

**Rating:** 6
**Confidence:** 3

**Review:**

The method shows promising results on on generating high duration (up to 100 frames) class conditional videos with convincing Inception scores, indicating quality similar to DVD-GAN, while consuming less memory and with better coherence.
While the contribution of the paper is mainly to improve the DVD-GAN architecture to reduce training time and memory consumption, the reviewer believes that the paper would be a good contribution to the venue.
Below there are some questions on the methodology:
1. Is there any way to tell does the matching discriminator actually only estimates the ability to upsample $\hat{x}_w$ from the previous low resolution sample $x_w^l$? From the architecture it is not evident whether or not it only does this or it is also entangled with assessment of how good the low-resolution sample $x_w^l$  was. In other words, if the low resolution sample scores good (e.g. because it's the real-world data) but the upsampling does not match,  would the objective of the matching descriptor training still score it as a good upscaling?  Or is there any reason preventing from this type of behaviour?
2. Although, as mentioned in the introduction, it may not be as big problem as for VAE-based models, the problem of blurring might exist for DVD-GAN-like models. It is written in the caption of Figure 5 that 'Despite the two stages of local upsampling, the frame quality does not degrade noticeably through time.’ Although the reviewer appreciates that previous work reported only IS/FID/FVD metrics and that defining proper evaluation metrics for generative models is an open question, it might be a good idea to show some other quantitative metrics such as power spectral density (PSD) plots similar to figure 5 from [1]. This would help get an idea how it compares to the real-world video in terms of blurring of the results.
3. Given that the generation of videos is class-conditional, is it possible to show the metrics per class? Are the scores per class similar or does the method score better for larger classes or the classes with specific motion dynamics?

[1] Ayzel et al (2020) RainNet v1.0: a convolutional neural network for radar-based precipitation nowcasting

---

> ### Author Response · Authors · 2020-11-14
> **Answer to AnonReviewer #1**
>
> Thank you for your review!. We address your questions below:
>
> 1. While the video discriminator in stage 1 and stage 2 only discriminate videos in isolation (x^w_l and x^w, respectively), the matching discriminator scores pairs (x^w_l, x^w). If the low-resolution sample x^w_l is good (for example, it is ground truth data) but the high resolution sample does not match, then the discriminator should be able to tell that fake pair apart from real pairs, pushing the second stage generator to produce a matching upscaling.
> In our theoretical analysis we show that, when discriminating pairs of (x^w_l, x_w), there exists a global minimum when the model and data joint distributions match.
>
> 2. We looked at the reference for the PSD metric, but we are unsure how to use it to assess our model, as in the reference there is a ground-truth video for a given prediction. Could you clarify how to use PSD to assess potential blurring in our generation setup? Nevertheless, we agree that the visual quality of the generation could be further improved. While we match state-of-art performance, generative modeling of videos is not a solved problem.
>
> 3. We randomly selected 5 classes with high motion/temporal variations and 5 classes with low motion and computed per-class scores. For each category we generated 1000 random samples and used all the available ground truth videos in that category (usually between 500 to 1000 examples) to compute the scores. We observe high variability in FID and IS across classes, but we do not observe significant trends within high motion or low motion classes. For example, there are categories in the non-motion category with FID score > 110 while some motion classes achieve a FID score of ~80 , and the IS scores are on average slightly better for the motion category. Looking at the samples, we also do not observe any significant distinctions between the different groups of classes.
> Instead we noticed differences in the generations across all classes in the dataset. We believe the class variability might be due to other factors such as amount of data available, having more semantically related classes or the amount of structure present in a particular class - generating simple objects is easier than generating faces.
> We will include this discussion with some visualizations in an updated manuscript.
>
> Below we provide the per-class scores we obtained:
>
> **Low Motion group**
>
> Category | FID | IS | # examples
> --------------------------
> Cooking_egg | 111.63 | 7.60 | 441
>
> Crying | 70.74 | 8.92 | 627
>
> Doing_nails | 91.67 | 13.32 | 537
>
> Reading_book | 64.97 | 11.13 | 793
>
> Yawning | 79.08 | 9.71 | 530
>
> **High Motion group**
>
>
> Category | FID | IS | # examples
> --------------------------
> Bungee_jumping | 82.21 | 11.66 | 799
>
> Capoeira | 84.57 | 9.48 | 816
>
> Cheerleading | 116.84 | 12.10 | 982
>
> Kitesurfing | 108.81 | 11.36 | 648
>
> Skydiving | 90.25 | 5.99 | 983
>
>
> Thanks again for your time. Aside from the PSD metric, we hope we have answered your remarks. Please let us know if you have further questions.

---

> > ### Comment · AnonReviewer1 · 2020-11-17
> > **Re: Answer to AnonReviewer #1**
> >
> > 1. The main difference between the provided theoretical analysis and the implementation, as the reviewer can see, would be  that it uses gradient descent (local) optimisation and therefore there could be some ways for the model to 'cheat' and get into a local minimum. Therefore, is it possible to visualise the examples of the original and upsampled version to show that it actually upsamples in the expected way?
> > 2. *We looked at the reference for the PSD metric, but we are unsure how to use it to assess our model, as in the reference there is a ground-truth video for a given prediction. Could you clarify how to use PSD to assess potential blurring in our generation setup? Nevertheless, we agree that the visual quality of the generation could be further improved. While we match state-of-art performance, generative modeling of videos is not a solved problem.*
> > Indeed, in Figure 4 of Ayzel et al (2020) they present PSD for the ground truth video for a given prediction. I was talking more about Figure 5 instead, which shows 'PSD averaged over all verification events and nowcasts'. It is my understanding that this should be possible to do without having directly corresponding ground truth for the predicted videos. This could give the idea of blurriness of the proposed methods' predictions vs real videos and DVD-GAN's. The motivation behind this comment was that the FID and IS metrics to assess the quality of predictions are very limited in their scope[1, 2], and it makes sense as evaluation of generative models is unsolved problem too. The question to the authors is whether they could see the way to mitigate the limitation of the analysis that it only outputs IS, FID and FVD scores. The reviewer knows it is a common place to use just these three scores, but thinks that it would be a nice thing to do as it would give some additional insight into the high-level characteristics of predictions.
> > Another thing which could improve the analysis would be to replicate the setup from the original DVD-GAN paper (BAIR Robot Pushing Dataset) on conditional frame prediction, but the reviewer also acknowledges that this setup could take more time than is available for the rebuttal. The benefit of evaluating on conditional frame prediction is that it enables a wide range of metrics (LPIPS, for example).
> > 3. Sounds interesting, looking forward to seeing the updated paper!
> > [1]Barratt, Shane, and Rishi Sharma. "A note on the inception score." arXiv preprint arXiv:1801.01973 (2018).
> > [2]Borji A. Pros and cons of gan evaluation measures. Computer Vision and Image Understanding. 2019 Feb 1;179:41-65.

---

> > > ### Author Response · Authors · 2020-11-19
> > > **Follow-up on comment by AnonReviewer #1**
> > >
> > > Thank you for your additional suggestions and helpful comments.
> > >
> > > 1. As requested, we have added a figure (Fig. 10 in the appendix F) that shows the second stage output along with the corresponding first stage output for some random samples from our model. The figure shows that the second stage refines the details of the first stage generation but preserves the overall scene structure. Please refer to appendix F for more details.
> > >
> > > 2. Thanks for your clarification. We have added plots that show the average PSD at different timesteps for multiple samples from the ground-truth data as well as our predictions. The plots show that our predictions have similar PSD to the ground truth, and that this is true even for the last frame of the generations. This aligns with our observation that our samples, while not necessarily accurate, do not significantly blur over time. Please refer to section E in the appendix for further details. Thank you for suggesting this experiment!
> > >
> > > 3. We have updated the paper following our previous comment with a discussion on the effect of class motion on the sample quality. We have included the table as well as a visualization of some of the samples generated for the high motion and low motion categories. This discussion can be found in section G of the appendix.
> > >
> > > Thank you again for your constructive suggestions. We hope that this discussion can help you confirm your favorable assessment of our submission. Please let us know if you have additional comments or questions.

---

> > > > ### Comment · AnonReviewer1 · 2020-11-24
> > > > **Re: Follow-up**
> > > >
> > > > The reviewer  checked the improvements on analysis, and the reviewer maintains the initial recommendation for acceptance. The reviewer thinks that their comments, as well as the comments of the other reviewers, have been addressed accordingly.
> > > >
> > > > The following are the reasons for the given score and recommending acceptance.
> > > >
> > > > Strengths:
> > > > -The paper is well written and produces an important new GAN-based baseline for the open research question of video prediction, addresses the problem of video prediction for up to 100 frames which could not be solved using the previous methods, and, what is also important, addresses the existing drawback of excessive time consumption for training the model.
> > > > - The analysis and ablation studies have been significantly improved
> > > >
> > > > Weaknesses:
> > > > - It is focused on improvements of the existing architecture (but the reviewer thinks that this is still a valid contribution as it addresses multiple limitations of the baseline architecture such as the duration of the predicted video and training time)
> > > > - Unlike the original DVD-GAN paper, this paper does not produce any results for the conditional setting which limits the comparison with the alternative methods such as DVD-GAN, but the reviewer accepts that it may not have been possible due to the time limits of the rebuttal ; however, this point was mitigated by improvement in presented additional experiments including those which haven't been presented in DVD-GAN paper (PSD)

---

### Official Review · AnonReviewer3 · 2020-10-28
**Official Blind Review #3**

**Rating:** 3
**Confidence:** 3

**Review:**

Summary:
The paper proposes a stage-wise training pipeline for training 128x128 resolution videos of up to 100 frames. It starts by generating low resolution and temporally downsampled videos, and upsample the results in a stage-wise manner. Experimental results on Kinetics-600 and BDD100K demonstrate that the network is effective in generating higher resolution videos.

Strengths:
The idea is easy to understand. The paper is well written and easy to follow. Quantitative results show that the proposed method is superior than existing methods under some circumstances.

Weaknesses:
1.	The novelty is very low. Stage-wise and progressive training have been proposed for such a long time, they have been used everywhere. The way the authors use them don’t really exhibit anything novel to me.
2.	The resolution of the outputs (128x128) is lower than prior works (e.g. DVD-GAN has 256x256 outputs). Since the paper claims the computation cost is lower, one would expect the model can generate higher resolution and much longer duration videos, but in fact it’s quite the opposite. To prove the effectiveness, I feel the authors need to show something higher than 256x256, say 512 or 1024 resolution. On the other hand, the hardware requirement is still high (128 GPUs) instead of some normal equipment that everyone can have, so I really don’t see any benefit of the model. If the authors can train DVD-GAN using only a handful of GPUs, that might also be a contribution, but it’s not the case now.
3.	Output quality is reasonable, but still far from realistic. Recent GAN works have shown amazing quality in synthesized results, and the bar has become much higher than a few years ago. In that aspect, I feel there’s still much room for improvement for the result quality.

Overall, given the limited novelty, low resolution output and still high hardware requirement, I’m inclined to reject the paper.

---

> ### Author Response · Authors · 2020-11-14
> **Answer to AnonReviewer #3**
>
> We thank you for your time and effort in reviewing our paper.
>
> 1. To the best of our knowledge, we are the first to propose a stage-wise generative model for video where **each stage is trained independently**. We show that our approach is competitive with DVD-GAN, which is the current state-of-the-art for video generation. Furthermore, we demonstrate that our approach can generate videos of up to 100 frames at 128x128 resolution. Our method is the first one to produce such generations. Finally, we show that approach is theoretically grounded.  We believe that these contributions are novel and significant. We also note that the fact that we have multiple stages is a byproduct of our motivation and formulation rather than the novelty of our paper. We would happily discuss the novelty in relation with some related works if you provide us with some references.
>
> 2. A comparison to previous video generation models in terms of output dimensionality should take into consideration the length of the videos. While the spatial resolution of our results is the same as previous approaches, to the best of our knowledge we are the first ones to generate 128x128 videos of up to 100 frames. Note that the dimensionality of 128x128x100 is considerably higher than the 256x256x12 results found in DVD-GAN.
> Furthermore, we are able to generate such videos while requiring 2x less computational resources, i.e. we could further increase the output dimensionality if using the same amount of computational resources as DVD-GAN. While the hardware requirement is still high, it is nonetheless 2x less than current state-of-art approaches.
>
> 3. We would like to point out that video generation is a hard task which is not solved. Video generative models are far from being at the quality level of their images counterpart.  It is therefore inappropriate to compare the quality of image-based generative models with video generative models.
> Our results are of similar or better quality to DVD-GAN according to the IS and FID metrics, which is the state-of-the-art in video generation. Our samples also are of similar qualitative quality than DVD-GAN.  For reference, these are some 128x128 generations from DVD-GAN, provided by the DVD-GAN authors: https://drive.google.com/file/d/1P8SsWEGP6tEGPPNPH-iVycOlN6vpIgE8/view.
> We would be happy to compare to methods that produce more realistic results on Kinetics if you  could provide us with a reference and a metric for the comparison.
>
> Again, thank you for your review. We hope that our answers clarify the significance and novelty of our contributions. We hope that you will reconsider our score, which we believe is severe with our work. Let us know if you have any further questions or concerns.

---

### Official Review · AnonReviewer5 · 2020-11-05

**Rating:** 7
**Confidence:** 5

**Review:**

**Paper Contributions**

The paper proposes SSW-GAN, an adversarial generative model of video which proposes a new generator architecture along with splitting the training into multiple stages.

**Strong points of the paper**

* The results are very strong.
* Prior work has focused on efficient decomposition of the discriminator, this work focuses on decomposition of the generator (effectively), and this is an extremely reasonable direction to take adversarial video research in.
* The claim that this requires substantially less computational cost is grounded.

**Weak points of the paper**

* A major departure from prior work is training stage 1 of SSW-GAN separate of stage 2, but this is independent of the computational benefits present from the generator architecture innovations. It is missing an important ablation showing the results of training the stages jointly.
* The generator architectures changes from prior work are quite concrete, but the high level description doesn't clearly reflect that.
* Some of the comparisons and model descriptions are lacking details, which make it difficult to understand experiments.

**Clearly state your recommendation (accept or reject) with one or two key reasons for this choice.**

I believe this paper is an accept (7), however I think there are a number of places where the description of the architecture and experiments needs more detail, and for my final rating I would like to see these addressed in rebuttal.

Furthermore, I think there is a key experiment missing. Were that to be added I would strongly consider moving to clear accept (8).

**Supporting arguments for your recommendation.**

There are two key innovations in the SSW-GAN model:
* A generator architecture which provides output to discriminators that does not require an entire high-resolution full-length sample to be generated (because between stage 1 and 2 you select only a window of frames to run stage 2 on).
* Splitting the training into two phases, where at first you only train stage 1 and then you train stage 2.

These are very interesting ideas, since either one substantially reduces the training-cost of the video model (something aptly described in the paper), and not something strongly touched on by previous work. In addition, the metric scores of the SSW-GAN model are very nice, state of the art in almost all cases. For that reason I think the work in this paper is worthy of acceptance.

However there are a couple points of clarification and increased description which I think are needed, and I would like to see the following points addressed in the rebuttal:

* The core idea of judging fixed-length output upscaling as a generative problem seems like a very clear and reasonable idea. Most of the high-level description of the most omits this in place of a general “multi stage definition” of the SSW-GAN model. I think the introduction and abstract would benefit about being more clear with regards to this change.

* In section 4, it is unclear if all four bold sections are trained independently or not. I think it would be good to be clear in the paper-structure which components are trained together.

* In general, the paper is not super clear where upsampling (both in time and space) occurs. I think it would be good to describe this in the paper text and in the architecture figures.

* I am not super clear on the comparisons between SSW-GAN and DVD-GAN in “comparison with prior work”. When you say "our model trained to generate 128x128/12 videos “ is this the model which had a first stage trained on 32s32/25 and then you trained the second stage on input windows of 6 frames, generating 128x128/12, and took just single samples from that to compare?

* Similarly, when you say “However, our model is only trained on 128x128/12 outputs, as it is unrolled and applied convolutionally over the first stage output to generate 48 frames.”, isn’t it the case that the first stage is trained on longer sequences?

*  In Table 1, the numbers for DVD-GAN seem lifted from the paper, which I believe is using a Kinetics-600 trained I3D for metric calculation. This means the FID numbers are comparable, but in section 5 when describing IS you say you are using a Kinetics-400 trained I3D (like in FVD). Is this correct (in which case the numbers are not quite comparable) or is this is just mis-explained, and all numbers in Table 1 come from Kinetics-600 trained I3Ds?

* I think the IS/FID metrics you pick in Table 1 are the better metrics, but could you also add FVD numbers? That would allow you to compare against TriVD-GAN [1], which outperforms DVD-GAN. I think this is important because that paper also discusses modifications which reduce the memory requirement of DVD-GAN.

Finally, there is a major question this paper does not address: **is the two-stage training of SSW-GAN necessary, or can it be trained in a single pass** (but with the generator decomposition as described)? Doing so would be simpler, and also might potentially reduce the need for the Matching Discriminator, which the paper contains an ablation for, but I think needs a further ablation when the model is not trained in two stages.

I believe a key experiment would be training an SSW-GAN architecture but in a single pass, the results of that experiment would mean quite a lot for interpreting the changes described in this paper. In particular, the paper's title and abstract focus on the multi-stage aspect of the model, but skim over the substantial change of making the generator architecture more modular and scalable. I think it would be very beneficial to understand how each of these changes independently effect the performance of the model.

I do not think this ablation is required to maintain an accept rating, but including it would push my rating closer to clear accept, and be quite a strong addition to the paper's content.

[1] https://arxiv.org/abs/2003.04035

---

> ### Author Response · Authors · 2020-11-14
> **Answer to AnonReviewer #5**
>
> Thank you for your time in writing the review for our paper. We really appreciate your very thoughtful and detailed review. We found your feedback very helpful, thanks!
>
> Below we address your questions:
>
> 1. We agree that the joint training of multiple stages is an exciting direction to explore. The main difficulty in training stages jointly is that we have to keep the activations of all previous stages in memory for backpropagation, increasing the overall memory requirements of the model compared to training stages independently. For the particular model we propose in this submission, this brings the memory requirements close to DVD-GAN.
> We are currently performing an initial experiment on a two-stage model on Kinetics in which all stages are trained jointly. The second stage takes a random window out of the first stage generation as input, and we directly discriminate on the output of the second stage without a matching discriminator. We observe two main things: first, not using a matching discriminator causes the output of stage 1 to not be a proper low resolution video. While it is still used by the second stage, there is nothing that encourages the output of stage 1 to be a real video. Second, this model seems to be more inconsistent than the equivalent SSW-GAN when unrolled, as the low resolution but complete first stage output is not grounded. Some of these issues could be addressed by still using a matching discriminator or a discriminator for the first stage output.
> Thank you for suggesting this experiment. We will add those results in the next revision of our paper once the training is fully complete.
>
> 2. The model is trained in stages, all components in a given stage are trained at the same time and independently of other stages. The first stage is trained separately of the upsampling stage, with the latter one including the conditional generator and the matching discriminator. We will update Section 4 to clarify this.
>
> 3. Upsampling is done differently for the temporal and spatial dimensions. Spatial upsampling is achieved through progressively increasing the resolution of the intermediate activations of the upsampling generator. On the other hand, temporal upsampling is done at the input of the upsampling generator, where the noise vector already has full temporal dimension, and we temporally upsample the low resolution video accordingly. This is similar to how DVD-GAN operates - the noise has full temporal size, while the spatial resolution is increased progressively in the generator. We will clarify this in the paper
>
> 4. All our first stages in the paper are trained to generate 32x32/25 videos. Then, as you mention, we train the second stage on 32x32/6 windows to generate 128x128/12 outputs. During evaluation, the scores obtained for 128x128/12 are obtained by upsampling 6 randomly selected frames from the 32x32/25 low-resolution videos. It is true that the first stage always generates 25 frames and we will clarify it in the paper.
> Stage 2 models can be unrolled over the full first stage output to generate 128x128/50 videos. We also include an ablation where we train on smaller 32x32/3 windows to generate 128x128/6 frames, but still unroll the model to generate 128x128/50 videos.
>
> 5. The explanation for how we compute the metrics is correct. We corresponded with the authors of TriVD-GAN[1], which overlap with authors of DVD-GAN, to make sure we used the same evaluation setup as DVD-GAN. We included this explanation because at first we were unable to reproduce their metrics. As you mention, all numbers in our experimental section for DVD-GAN are taken from their reported results.
>
> 6. We will compute and report FVD metrics. Note however that TriVD-GAN focuses on video prediction, and that DVD-GAN only reports FVD metrics for the BAIR dataset and their video prediction results on Kinetics, while we focus on video generation.
>
> Again, we thank you for your insightful comments and suggestions. We will soon update the paper to clarify some of the points you raised including details about the experiments, the architectural details and the overall training scheme, as well as to more clearly state the main idea. Please let us know if you have any other questions.

---

> > ### Author Response · Authors · 2020-11-25
> > **Follow-up**
> >
> > We have now updated the paper to include FVD scores for our models for the Kinetics and BDD experiments.
> >
> > Thank you again for your review. We hope we have addressed your concerns and confirmed your positive review of our submission with the updated revision.

---

### Author Response · Authors · 2020-11-19
**Paper Update**

Dear reviewers and area chair,

First we would like to thank you for your thoughtful reviews. We were happy to hear that the paper shows competitive results on real world datasets [R1, R4, R5], that it is exploring a reasonable research direction for adversarial video generation [R5] and that it would constitute a good contribution to the venue [R1, R5].

We want to point out the following changes that we did to the paper thanks to your feedback:

* **[R1]** We added a comparison looking at the Power Spectrum Density (PSD) between the original data and our generations in Appendix E. We look at the PSD for different timesteps: 1, 10, 25, 50. We observe that our generations have a similar PSD to that of the original data, even at the end of the generation, indicating that the generations do not blur significantly and remain stable over time. We also added a citation to (Ayzel et al., 2020). Thank you for suggesting this experiment!

* **[R1, R4]** We added an additional analysis looking at the difference between classes with high motion versus classes with low motion in Appendix G.  We randomly selected 5 classes with high motion/temporal variations and 5 classes with low motion and computed per-class scores. For each category we generated 1000 random samples and used all the available ground truth videos in that category (usually between 500 to 1000 examples). We compute FID/IS scores and observe high variability in FID and IS across classes, but we do not observe significant trends within high motion or low motion classes. We also added additional samples from those classes. Looking at the samples, we also do not observe any significant distinctions between the different groups of classes.

* **[R1]** We provide an additional upsampling visualization in Appendix F showing that the upsampling stage is well grounded in the low-resolution videos as expected.

* **[R5]** We updated the text to clarify the architecture details and experimental setting. Please let us know if this addresses your concerns.

* **[R5]** We included initial results on an experiment where we train a two-stage model end-to-end. We train this model without a first stage discriminator nor a matching discriminator, instead directly discriminating on the second stage output. We observe that the output of the first stage is no longer a valid low resolution video, but the second stage generates valid snippets nonetheless. We also empirically observe that samples generated by the single pass end-to-end model do not show a lot of motion. Please refer to Appendix H for more details.

* **[R4]** We added the provided citations. Thank you for pointing them out!

Thank you again for taking the time to review our paper. We hope that this will address your concerns, please let us know if you have further questions.

---

> ### Author Response · Authors · 2020-11-25
> **Paper Update #2**
>
> We have now uploaded a second revision of our paper.
>
> * **[R1]** We have now added FVD scores for our models for the Kinetics and BDD experiments. We hope this will ease future comparisons to our method.

---

### Decision · Program_Chairs · 2021-01-07
**Final Decision**

**Decision:**

Reject

**Comment:**

This paper proposes a GAN for video generation based on stagewise training over different resolutions, addressing scalability issues with previous approaches. Reviewers noted that the paper is clearly written, proposes a method that improves upon the DVD-GAN architecture by reducing training time and memory consumption, and has competitive quantitative results.

On the other hand, the more negative reviewers are concerned that the empirical improvements demonstrated are somewhat incremental, and that there is not much novelty as the proposed approach is similar to other methods that decompose the generation process into multiple stages at different temporal window lengths and/or spatial resolutions. The authors argue that these criticisms are subjective and non-actionable. I sympathize with their frustration, but an acceptance decision for a competitive conference like ICLR does involve some subjective judgment as to whether the method and/or results meet a high bar beyond mere correctness. For this submission that's a close call, but between the novelty/incrementality concerns and the other more minor issues raised by reviewers (e.g., missing frame-conditional evaluation) I believe this paper could benefit from another round of revisions and improvements and recommend rejection.

I hope the authors will consider improving the submission based on the reviewers' feedback and resubmitting to a future venue, as the paper certainly has merit. To this end I have a few concrete recommendations for the authors which could have flipped my recommendation to an accept if implemented:

* Report results in the frame-conditional setting for comparison with DVD-GAN and other methods that operate in this setting.
* Proofread the paper more thoroughly. I noticed several typos while skimming the paper, e.g. in the theory section, the second term of eq. 6 confusingly uses $\rho$ instead of $\log$. (Relatedly, given that appendix B.1 reports that the hinge loss is used, I'm not sure whether $\log$ is correct in the first place -- this probably deserves further explanation or correction.)
* Demonstrate/argue more convincingly (in one way or another) that SSW-GAN's improved efficiency really expands the frontier of what was possible before. It is true that the 128x128/100 video samples contain 2x as many total pixels as DVD-GAN's 256x256/12 samples, but this isn't a *strict* improvement as the spatial resolution is smaller, and a 2x difference leaves space for reviewers to reasonably wonder whether previous methods really couldn't have matched this if pushed. Some possible examples of this: show that SSW-GAN can generate longer 256x256 videos (a strict improvement over what was possible with DVD-GAN), or orders of magnitude longer (e.g., 1 minute) but still temporally coherent videos at 128x128, or videos with substantially improved subjective sample quality at the same (or higher) resolution.
* The paper notes that "DVD-GAN models do not unroll well and tend to produce samples that become motionless past its training horizon". If this were quantified, e.g. by additionally reporting IS/FID/FVD separately for different timestep ranges, it could make a more compelling argument in favor of SSW-GAN.